# Atypical Hepatitis B Virus Serology Profile—Hepatitis B Surface Antigen-Positive/Hepatitis B Core Antibody-Negative—In Hepatitis B Virus/HIV Coinfected Individuals in Botswana

**DOI:** 10.3390/v15071544

**Published:** 2023-07-13

**Authors:** Bonolo B. Phinius, Motswedi Anderson, Margaret Mokomane, Irene Gobe, Wonderful T. Choga, Tsholofelo Ratsoma, Basetsana Phakedi, Gorata Mpebe, Doreen Ditshwanelo, Rosemary Musonda, Joseph Makhema, Sikhulile Moyo, Simani Gaseitsiwe

**Affiliations:** 1Botswana Harvard AIDS Institute Partnership, Private Bag BO 320, Gaborone, Botswana; bphinius@bhp.org.bw (B.B.P.); manderson@bhp.org.bw (M.A.); wchoga@bhp.org.bw (W.T.C.); tratsoma@bhp.org.bw (T.R.); bphakedi@bhp.org.bw (B.P.); gmpebe@bhp.org.bw (G.M.); dditshwanelo@bhp.org.bw (D.D.); rmusonda@bhp.org.bw (R.M.); jmakhema@bhp.org.bw (J.M.); smoyo@bhp.org.bw (S.M.); 2School of Allied Health Professions, Faculty of Health Sciences, University of Botswana, Private Bag UB 0022, Gaborone, Botswana; mokomanem@ub.ac.bw (M.M.); gobei@ub.ac.bw (I.G.); 3Department of Immunology and Infectious Diseases, Harvard T. H. Chan School of Public Health, Boston, MA 02115, USA; 4School of Health Systems and Public Health, University of Pretoria, Private Bag X20, Hatfield, Pretoria 0028, South Africa

**Keywords:** hepatitis B surface antigen (HBsAg), hepatitis B core antibodies (anti-HBc), HBV, HIV, Botswana, Africa

## Abstract

(1) Background: Hepatitis B core antibodies (anti-HBc) are a marker of hepatitis B virus (HBV) exposure; hence, a normal HBV serology profile is characterized by HBV surface antigen (HBsAg) and anti-HBc positivity. However, atypical HBV serologies occur, and we aimed to determine the prevalence of an atypical profile (HBsAg+/anti-HBc-) in a cohort of people with HIV-1 (PWH) in Botswana. (2) Methods: Plasma samples from an HIV-1 cohort in Botswana (2013–2018) were used. The samples were screened for HBsAg and anti-HBc. Next-generation sequencing was performed using the GridION platform. The Wilcoxon rank-sum test and Chi-squared tests were used for the comparison of continuous and categorical variables, respectively. (3) Results: HBsAg+/anti-HBc- prevalence was 13.7% (95% CI 10.1–18.4) (36/263). HBsAg+/anti-HBc- participants were significantly younger (*p* < 0.001), female (*p* = 0.02) and ART-naïve (*p* = 0.04) and had a detectable HIV viral load (*p* = 0.02). There was no statistically significant difference in the number of mutations observed in participants with HBsAg+/anti-HBc- vs. those with HBsAg+/anti-HBc+ serology. (4) Conclusions: We report a high HBsAg+/anti-HBc- atypical serology profile prevalence among PWH in Botswana. We caution against HBV-testing algorithms that consider only anti-HBc+ samples for HBsAg testing, as they are likely to underestimate HBV prevalence. Studies to elucidate the mechanisms and implications of this profile are warranted.

## 1. Introduction

Approximately 296 million people were reported to be living with chronic hepatitis B (CHB) infections in 2019 [1]. The hepatitis B virus (HBV) is one of the leading causes of liver-related mortality, with up to 820,000 deaths annually [1]. Worse-off clinical outcomes may be more pronounced in people with concomitant HBV/human immunodeficiency virus (HIV) than in those with either mono-infection [2,3]. Botswana has an HBV prevalence ranging from 1.1% to 10.6% in different populations [3,4,5,6], with an HBV incidence of 3.6/100 person-years in people with HIV (PWH) [7]. HIV prevalence in Botswana is 20.8%, and the country has a successful national HIV management program that has resulted in Botswana surpassing the UNAIDS 95-95-95 goals [8]. 

The HBV genome is partially double-stranded with four overlapping open reading frames (ORFs), namely the polymerase, which codes for the reverse transcriptase enzyme; the *precore/core (preC/C)*, coding for the hepatitis B e antigen (HBeAg) and the hepatitis B core antigen (HBcAg), respectively; the *preS1/preS2/S*, coding for the small, middle and large surface antigens; and the *X* gene for the transcriptional trans-activator protein [9]. HBV biomarkers such as the hepatitis B surface antigen (HBsAg), HBeAg and HBV deoxyribonucleic acid (DNA) are used for diagnosis and disease monitoring. These biomarkers are produced at different levels throughout the course of an HBV infection [10], and some are detected using serological assays. Typically, HBsAg, which is a diagnostic marker, can be detected 4–10 weeks after infection, and the detection of this antigen in the blood within 6 months of infection is deemed an acute infection, while its persistence after 6 months is deemed a chronic infection. Anti-HBc immunoglobulin M (IgM) antibodies are detected early on because HBcAg is highly immunogenic [11,12]. As anti-HBc IgM levels decline, anti-HBc immunoglobulin G (IgG) predominates and normally persists for life, indicating exposure to HBV infection [11]. Anti-HBc levels differ at different stages of HBV infection, normally being high during active infection [13]. 

Anti-HBc positivity is commonly used to select participants to screen for HBsAg in HBV surveillance studies [14,15]. Anti-HBc loss has been reported in CHB patients [16,17], thereby resulting in an atypical HBV serology profile characterized by HBsAg-positive serology in the absence of anti-HBc (HBsAg+/anti-HBc-). In one patient, a permanent loss of anti-HBc was observed during an HBV reactivation episode after an allogenic stem-cell transplantation, which was discussed as being due to the depletion of the recipient’s anti-HBc-specific B cell clones [18]. This serology profile has also been observed during an acute HBV infection in a non-Hodgkin lymphoma patient [19]. Several studies attribute this serology profile to immune suppression [16,17,19], while one case report attributes it to mutations in the core region of the HBV genome [20]. The sensitivity and specificity of the serological assays used could also result in false negatives, as shown by one study that had 17% of samples becoming detectable for anti-HBc with a different kit after initially testing negative with a less sensitive kit [21]. Other atypical HBV serology profiles and mechanisms behind them have been reviewed elsewhere [22]. A few studies have been undertaken to describe the HBsAg+/anti-HBc- atypical profile in sub-Saharan Africa [23], but none in Botswana. We aimed to determine the prevalence of this atypical HBV serology profile (HBsAg+/anti-HBc-) and its associated factors in a large cohort of PWH in Botswana. 

## 2. Materials and Methods

### 2.1. Study Population

In this retrospective study, stored plasma samples of PWH who participated in the Botswana Combination Prevention Project (BCPP) (2013–2018) were used. The BCPP study enrolled 12,610 consenting participants with the main aim of evaluating the impact of a combination prevention package on HIV incidence [24]. A random sample of 20% of adults aged 16–64 years was enrolled from 15 paired communities. These communities were matched by size, pre-existing health services, population age structure and geographic location. Of the 12,610 participants, 3596 were PWH [24]. This study was approved by the University of Botswana Institutional Review Board (UBR/RES/URB/BIO/267) and the Health Research and Development Committee (HRDC) at the Botswana Ministry of Health (HPDME 13/18/1).

### 2.2. Serology Screening and Viral Load Quantification

Participant plasma samples that were previously screened for various HBV serology markers were used [6]. Briefly, plasma samples were initially screened for HBsAg (Murex Version 2, Diasorin, Dartford, UK) and total core antibodies (anti-HBc) using the Monolisa anti-HBc PLUS ELISA kit (Bio-Rad, Marnes-la-Coquette, France). Samples with positive HBsAg serology (HBsAg+) were then screened for HBeAg and anti-HBc IgM using the Monolisa HBe Ag/Ab and Monolisa anti-HBc Plus 1 Plaque (Bio-Rad, Marnes-la-Coquette, France), respectively. Following the manufacturer’s instructions, HBV DNA was quantified using the COBAS AmpliPrep/COBAS TaqMan HBV Test version 2.0 (Roche Diagnostics, Mannheim, Germany), which has a broad linear range of 20 IU/mL–1.7 × 10^8^ IU/mL.

### 2.3. Sequencing Methods

The QIAamp DNA Blood Mini kit (Qiagen, Hilden, Germany) was used to extract DNA from plasma according to manufacturer’s instructions with a final elution volume of 30 μL (Qiagen, Hilden, Germany). A two-step PCR was performed to amplify the whole HBV genome. Briefly, master mixes were prepared for two primer pools. The first-round master mixes were composed of 1.5 μL of nuclease-free water, 0.05 μL of primer pool, 6.25 μL of Q5^®^ Hot Start High Fidelity 2× master mix (New England Biolabs, Ipswich, MA, USA) and 5 μL of template volume. PCR conditions were: an initial denaturation at 98 °C for 30 s, 35 cycles of a denaturation step at 98 °C for 15 s, annealing and extension steps at 65 °C for 5 min and, finally, a hold step at 4 °C for ∞. Second-round PCR master mixes for each primer pool were composed of 0.5 μL of the primer pool, 3.75 μL of nuclease-free water, 6.25 μL of the Q5^®^ Hot Start High Fidelity 2× master mix and 2.5 μL of the template. First-round PCR conditions were used for second-round PCR. Library preparation was carried out as per the Oxford Nanopore PCR tiling with rapid barcoding and midnight expansion protocol (version MRT_9127_v110_revH_14Jul2021) [25], replacing SARS-CoV-2 primers with HBV primers. 

The library was loaded into flow cells in version R9.4.1 (Oxford Nanopore Technologies, Oxford, UK). Sequencing was performed using the GridION platform (Oxford Nanopore Technologies, Oxford, UK), with base calling and demultiplexing carried out using the latest MinKNOW Release as of April 2023 in high-accuracy mode. Raw FASTQ files were exported and subsequently processed using Guppy, employing dual-indexed reads for base calling and demultiplexing. For reference-based assembly of HBV, we utilized Genome Detective 1.132/1.133 [26] (https://www.genomedetective.com/, last accessed 15 April 2023). 

### 2.4. Sequence Diversity and Mutational Analysis

The generated HBV sequences were viewed and aligned using AliView version 1.26 [27]. Geno2pheno (https://hbv.geno2pheno.org, accessed 17 April 2023) was used to determine HBV subgenotypes. Babylon Translator was utilized to extract the core ORF, and the sequences were translated into amino acids [28]. We analyzed for core mutations per identified HBV subgenotype in our study. For subgenotype A1, 80 sequences were analyzed, including 10 HBsAg+/anti-HBc- from our study and 70 HBsAg+/anti-HBc+ from our current study and previous Botswana sequences from GenBank [29]. We only had one subgenotype D2 and one subgenotype E HBsAg+/anti-HBc- sequence; therefore, for subgenotypes D2 and E, we analyzed for mutations that have been reported among anti-HBc-negative participants elsewhere [21].

### 2.5. Statistical Analyses

We calculated the Wilson 95% confidence interval (95% CI) around prevalence estimates. Continuous variables were presented in medians and interquartile ranges and compared between atypical profile carriers (HBsAg+/anti-HBc-) and normal profile carriers (HBsAg+/anti-HBc+) using the Wilcoxon rank-sum test. Categorical variables were presented in proportions, and the Chi-squared test was used to compare variables between atypical profile carriers and normal profile carriers. A proportion test was performed to determine the association between HBV serology profile and sex, HIV suppression and ART status. A Fisher’s exact test was used to determine the association between presence of mutations and HBV serology profile type. All statistical analyses were performed in Stata version 18.0 (StataCorp LLC, College Station, TX, USA), and *p*-values lower than 0.05 were considered statistically significant. 

## 3. Results

### 3.1. Atypical Profile Prevalence and Participant Demographics

A total of 263 participants were screened for both HBsAg and anti-HBc. Most participants were female (63.1%, 166/263), and their median age was 42 (IQR: 35–50). Most of the participants had an undetectable HIV viral load (VL) (<40 copies/mL) (211/262, 80.5%). Most participants were on antiretroviral therapy (ART) (229/260, 88.1%), with 92/163 (56.4%) on a tenofovir disoproxil fumarate (TDF)-containing regimen. A total of 36 (13.7%, 95% CI 10.1–18.4) of the 263 participants were HBsAg+/anti-HBc-. There was an association between sex and the serology profile (*p* = 0.02) (Table 1). With a proportion test, it was determined that the prevalence of this profile was significantly higher in females than males (17.5% vs. 7.2%, *p* = 0.02). Participants with the atypical profile were significantly younger than those with the normal serology profile (35 (30–44) vs. 43 (36–50), *p* < 0.001). The prevalence of the atypical profile was higher among ART-naïve participants than ART-experienced participants (25.8% vs. 12.2%, *p* = 0.04). Furthermore, there was an association between the serology profile and HIV VL (*p* = 0.02) (Table 1), with a high prevalence observed among participants with detectable VL as opposed to those with undetectable VL (23.5% vs. 11.4%, *p* = 0.02). There was no statistically significant difference between participants with HBsAg+/anti-HBc- and those who had HBsAg+/anti-HBc+ serology profiles in terms of nadir CD4+ T-cell count, ART regimen or duration on ART (Table 1). The majority of participants (80% being HBsAg+/anti-HBc-, 85.4% being HBsAg+/anti-HBc+) on a TDF-containing regimen had an HBV viral load of less than 2000 IU/mL, including those with undetectable HBV viral load (Appendix A).

### 3.2. Sequencing Results 

The subgenotypes identified among participants with HBsAg+/anti-HBc- serology profiles were A1 (11/14 (78.6%)), E (2/14 (14.3%)) and D2 (1/14 (7.1%)). Ten atypical profile subgenotype A1 sequences had core region coverage; one subgenotype D2 and one genotype E also had core region coverage. Mutations were identified in 4/10 (40.0%) HBsAg+/anti-HBc- A1 sequences (Figure 1) vs. 27/70 (38.6%) HBsAg+/anti-HBc+ of the same subgenotype. There was no statistical significance in the number of mutations observed in participants with and without the atypical serology profile (*p* = 1.00). 

Mutations were identified at a frequency of one for most participants; one participant had two mutations (BBP089), while one participant with subgenotype A1 had nine mutations in the core region (BBP140). Mutations were observed from positions 42–182 of the core region (Table 2). 

## 4. Discussion

The prevalence of the atypical serology profile (HBsAg+/anti-HBc-) in our study was 13.7%. This serology profile was observed in 10% of PWH in Brazil [30], 4.3% of PWH initiating ART in Zimbabwe [23], and a much lower percentage (1.8%) in France [16]. These results show that HBV-prevalence-screening algorithms that only test HBsAg among individuals with positive anti-HBc serology [14,15] may underestimate HBV prevalence. Participants with the atypical profile in our study were significantly younger and female. A previous study showed a higher anti-HBc prevalence in males vs. females [31], while another, more recent study showed no statistically significant difference in anti-HBc levels between young male and female blood donors [32]. Other studies showed higher anti-HBc prevalence in older participants than younger participants [33,34]. Anti-HBc production may therefore be inherently lower in females and younger participants, as observed in these studies and our study. Highly sensitive quantitative anti-HBc assays may be useful in determining anti-HBc titer levels by age and gender. 

HBV subgenotypes A1, D2 and E were observed among atypical serology profile participants. These genotypes have been shown to be circulating in Botswana [4,29]. Several mechanisms have been put forth in describing the cause of this atypical serology profile. The lack of anti-HBc detection is expected in vaccinated individuals, who would normally test negative for HBsAg and positive for hepatitis B surface antibodies (anti-HBs) [35]. However, all our study participants were HBsAg positive, and we could not test for anti-HBs due to a lack of sample volume. However, an HBsAg+/anti-HBs+/anti-HBc- serology profile is unlikely, as per our previous study, in which none of the participants presented with this serology [36]. This might be because infant HBV vaccination started around 2000 in Botswana, and only one participant falls within this age range in the current cohort. 

Diagnostic escape mutations (G2011A, A2092T, G2138A, C2139T, C2242T and A2320G, with corresponding amino acid positions 37, 64, 80, 114 and 140) in the core region and immunotolerance of the core antigen have been put forth as some of the causes for this profile [20]. None of these mutations were identified in our study; however, we present other mutations in the core region as shown in our results. Specifically, mutations S141L/P and G123E are in the cytotoxic T-lymphocyte (CTL) epitope, while N74R, N75K, S87N and V74N are on the B-cell epitope [21]. Mutations in these epitopes are likely to affect core antigenicity, as reviewed elsewhere [37], which may explain the lack of anti-HBc detection in our study participants. Pre-exposure to HBV antigens in utero in infants born to HBeAg mothers has been said to cause an immunotolerance to HBeAg and HBcAg, resulting in anti-HBc negativity [38]. In a case study of a 62-year-old non-Hodgkin lymphoma patient who had recently received immunosuppressive therapy, this presentation was also observed and associated with a delayed antibody response due to her being immunocompromised [19]. The lack of antibody production due to immunosuppression caused by HIV has also been discussed [16,17,23]. In our study, on the other hand, participants with the atypical serology profile were no more immunocompromised than those with the normal serology profile. 

All our study participants were PWH, mostly on ART, and those participants who were specifically on TDF-containing regimens had undetectable HBV viral loads. While this undetectable viral load could be due to ART, a previous study from our group showed that 32% of HBV/HIV ART-naïve participants had an undetectable viral load, and an additional 18% had HBV viral loads of <20 IU/mL [3]. In the current study specifically, participants with an atypical serology profile were mostly ART-naïve with a detectable HIV VL, which may be an indication of declining CD4+ T-cell count even though a difference in CD4+ T-cell count between participants with an atypical serology profile and those with a normal profile was not observed. 

We acknowledge a few limitations in our study. We used the Monolisa anti-HBc PLUS ELISA kit (Bio-Rad, Marnes-la-Coquette, France) with a sensitivity of 99.5% and specificity of 99.5% and did not confirm with a different kit. With the high specificity of the assay, it is unlikely that the assay falsely identified anti-HBc-negative participants; however, there is room to query that. Studies that confirmed anti-HBc-negative results with a different kit showed some of these cases testing positive with a second kit (17% in one study [21] and 37% in another [35]). Anti-HBc levels are associated with HBV infection stage, with participants in the immune clearance phase and having HBeAg-negative hepatitis having higher median anti-HBc levels than those in the immune tolerance and low-replicative phases [13]. We could not ascertain infection stages in our study as most participants had been on ART for several years. We did not test for several other HBV markers, such as anti-HBs, due to sample volume issues, and we did not have liver enzyme test results for our study participants. We did not have vaccination records for the participants either, but it is unlikely that most participants were vaccinated as universal infant vaccination in Botswana was only implemented in the early 2000s [39]. Only one participant was 17 years old. Our study was focused only on PWH, which limits a comparative analysis between PWH and HIV-negative participants.

## 5. Conclusions

We report, for the first time in Botswana, participants with an atypical serology profile characterized by positive HBsAg and negative anti-HBc serology at a high prevalence of 13.7%. Participants with this profile were younger, female and ART-naïve with a detectable HIV VL. We caution against testing algorithms that only screen HBsAg among participants with positive anti-HBc serology, as they may underestimate the HBV burden, therefore missing out on participants in need of care. Future studies to elucidate the mechanisms and implications of this profile are warranted.

## Figures and Tables

**Figure 1 viruses-15-01544-f001:**
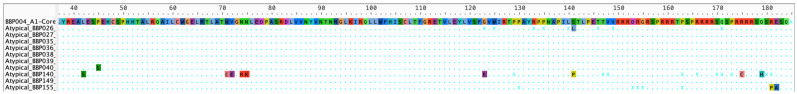
Subgenotype A1 amino acid sequence alignment of atypical serology profile sequences with a normal serology profile sequence (from position 37).

**Table 1 viruses-15-01544-t001:** Participant demographics.

Characteristics	(HBsAg+/Anti-HBc-)	(HBsAg+/Anti-HBc+)	*p*-Value
	N = 36	N = 227	
Sex, N (%)			
Female	29 (81)	137 (60)	
Male	7 (19)	90 (40)	0.02
Age in years, median (IQR)	35 (30–44)	43 (36–50)	<0.001
Nadir CD4, cells/mm^3^, n = 113, N (%)			
<200	8 (47)	34 (35)	
≥200	9 (53)	62 (65)	0.36
HIV VL			
Undetectable	24 (67)	187 (83)	
Detectable	12 (33)	39 (7)	0.02
ART status, n = 260, N (%)			
ART-naïve	8 (22)	23 (10)	
On ART	28 (78)	201 (90)	0.04
ART duration in years, n = 196, median (IQR)	7.0 (3.8–11.3)	6.6 (4.6–9.5)	0.79
ART regimen, n = 163, N (%)			
TFV-containing	11 (65)	81 (56)	
3TC-containing	5 (29)	59 (50)	
Non-TDF, non-3TC-containing	1 (6)	6 (4)	0.67
HBeAg, n = 231, N (%)			
Positive	2 (7)	24 (12)	
Negative	27 (93)	178 (88)	0.43
IgM, n = 244, N (%)			
Positive	3 (10)	13 (6)	
Negative	28 (90)	200 (94)	0.45
HBV VL IU/mL, n = 144, N (%)			
TND	8 (44)	29 (23)	
≤2000	6 (33)	72 (57)	
>2000	4 (22)	25 (20)	0.11

HBsAg—hepatitis B virus surface antigen; anti-HBc—hepatitis B virus core antibodies; HIV—human immunodeficiency virus; VL—viral load; ART—antiretroviral therapy; HBeAg—hepatitis B virus e antigen; IgM—immunoglobulin M; 3TC—lamivudine; TDF—tenofovir disoproxil fumarate; TND—target not detectable.

**Table 2 viruses-15-01544-t002:** Mutations identified in core region of HBsAg+/anti-HBc- participants.

Participant	Genotype	Mutations
Atypical_BBP026	A1	S141L
Atypical_BBP040	A1	P45S
Atypical_BBP140	A1	L42S, W71C, V72E, N74R, N75K, G123E, S141P, R175C, Q179H
Atypical_BBP149	A1	R181P, E182A
Atypical_BBP091	D2	S87N
Atypical_BBP089	E	D40E, V74N

## Data Availability

The data presented in this study are available upon request from the corresponding author. The data are not publicly available as the sequences are currently being analyzed for other objectives of the bigger project.

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
