# Peer review of "Atypical Hepatitis B Virus Serology Profile—Hepatitis B Surface Antigen-Positive/Hepatitis B Core Antibody-Negative—In Hepatitis B Virus/HIV Coinfected Individuals in Botswana"

_viruses, 2023, doi:10.3390/v15071544_

Round 1

Reviewer 1 Report

Comments and Suggestions for Authors

The manuscript under review focuses on HBV serology and sequencing in anti-HBc negative/HBsAg positive HIV infected patients. The manuscript is well written and the subject of the study is not novel, but has clinical significance.

Numerous studies in the past have described this phenomenon, very few, though, contained an analysis of HBV sequences. Reading the abstract, I was intrigued to learn more about the sequences analysis in the manuscript and was disappointed to see that this kind of analysis is largely missing. The authors state "Genetic diversity and any signature mutations were investigated by comparing the sequences isolated from patients with HBsAg+/anti-HBc- versus those with HBsAg+/anti-HBc+ serology". The following points should be addressed before publication:

a) How did the authors assess genetic diversity? How did the groups compare? Were the sequences compared to the reference subgenotypes? For example in a 2017 study (https://pubmed.ncbi.nlm.nih.gov/28622640/) the authors found greater variability in the core region of anti-HBc+ sequences, especially in a CD4+ epitope.

b) What would be the definition of signature mutation? Did the authors look at coding vs silent mutations? Which mutations/polymorphisms did they find in the 14 mutations? Did they look at minority variants as well?

Reviewer 2 Report

Comments and Suggestions for Authors

The manuscript by Phinius et al reports a limited study of HBsAg and anti-HBc profiles in HIV-1 infected patients, most receiving active ART. The percentage of cases negative for anti-HBc is puzzling but the design of the study and the partial data presented do not allow to identify any potential origin to this phenomenon.

Introduction. The last paragraph should present the range of hypotheses underlying the lack of anti-HBc detection: immunosuppression or immunodeficiency, genetic absence or change of critical core antigen, capture test core antigen lacking critical antigen, chronic HBV infection with anti-HBs only etc.

Methods. One weakness of the study is lack of testing for anti-HBs as a marker described in anti-HBs only chronic infection and indicating HBV vaccination that might also be involved in lack of anti-HBc.

Another flaw in the study design was the absence of comparison between an HIV-1-infected and non-infected group of CHB with or without detectable anti-HBc. Considering that immunodeficiency and lack of detectable anti-HBc was a major hypothesis, such comparison should have been done.

Results. In Table 1, in both anti-HBc pos and neg, patients with and without tenofovir should be stratified. The very high percentage of HBsAg pos patients with undetectable HBV VL might be related to ART cocktail. Patient with or without HBV vaccination should also be stratified (if any considering patient age distribution).

Regarding the issue of sex and HBV profile, a recent article should be quoted: Allain JP et al. Viruses 2022; 14: 673.

In sequencing data, the authors should focus on core protein sequences that might play a role if critical epitopes were different between anti-HBc pos and neg patients or between genotype A1 (dominant in Botswana) and A2 used to raise anti-c capture antigens.

Discussion. Although the initial observation is of interest, the design of the study does not provide the data enabling to examine various explanations such as: role of immunodeficiency (by comparing anti-c in HIV infected and non-infected cohorts HBsAg positive), the potential impact of ART cocktail with or without Tenofovir or other HBV active drugs, the potential role of HBV vaccination, impact of potential differences in core epitopes between genotypes A1 and A2 or between anti-HBc pos and neg patients.

Comments on the Quality of English Language

No major issue with language. a few typos to be corrected.

Round 2

Reviewer 1 Report

Comments and Suggestions for Authors

None. The points have been addressed.

Author Response

Comment: None. The points have been addressed

Response: Thank you for the important insights in the previous round of comments.

Reviewer 2 Report

Comments and Suggestions for Authors

The revised manuscript by Phinius et al was slightly improved but did not really address the main criticisms that should have requested additional testing, particularly for anti-HBs.

The issues of anti-HBs without anti-HBc reported in the literature was not addressed.

The specific analyses of cases HIV-1 infected versus non-infected; cases HIV-1 infected receiving Tenofovir or not, were not conducted and reported in results and discussion.

Core sequencing results essentially shows 6 cases wild type and only 1 case with multiple amino acid substitutions that might have influence anti-HBc detection. This analysis is not properly done, or its potential meaning discussed. The result being that core mutations is not a factor explaining lack of anti-HBc detection.

Table 2 is meaningless unless the substitutions are clearly related to each case showing that 9 of 16 are from a single case and none is common to more than 1 case.

The discussion should go far beyond just acknowledging the high frequency of no detectable anti-HBc and list potential causes and refuting or supporting each of them provided adequate analyses are done i.e. role of sequence in testing for anti-HBc, role of tenofovir in decrease of HBV viral load, comparing HIV-1 pos and neg cases etc.

Comments on the Quality of English Language

adequate
